# Making *Noah’s Ark* Work for Fishing Cat Conservation: A Blueprint for Connecting Populations across an Interactive Wild Ex Situ Spectrum

**DOI:** 10.3390/ani14192770

**Published:** 2024-09-25

**Authors:** Tiasa Adhya, Simran Singh, Himaja Varma Gottumukkala, Aditya Banerjee, Ishita Chongder, Sulata Maity, P. Anuradha Reddy

**Affiliations:** 1The Fishing Cat Project, Fishing Cat Conservation Alliance, P.O. Box 1488, Navasota, TX 77868, USA; adhyatiasa@yahoo.com (T.A.); himajavarma7475@gmail.com (H.V.G.); 2CSIR-Centre for Cellular and Molecular Biology, Uppal Road, Hyderabad 500 007, India; simran.suresh.singh@gmail.com; 3Human and Environment Alliance League, Bosepukur Road, Kolkata 700 042, India; adiban22@gmail.com; 4West Bengal Zoo Authority, Kolkata 700 106, India; ishita.titir@gmail.com (I.C.); maitysulata@yahoo.co.in (S.M.)

**Keywords:** captive breeding, wild cat, One Plan Approach, Opportunities to Thrive, species management, threatened

## Abstract

**Simple Summary:**

This study presents an integrated conservation framework for the globally threatened fishing cat, synthesizing both ex situ management and in situ strategies. The strategy aligns with the One Plan Approach, which views captive populations as part of a broader metapopulation network, and the Opportunities to Thrive framework, which emphasizes animal welfare in captivity. This study analyzed the genetic diversity of a recently initiated fishing cat captive breeding population by the West Bengal Zoo Authority, India, and conducted a population viability analysis. The results suggest that the current genetic diversity (56%) and population carrying capacity (30 individuals) can be sustained for over 100 years through a strategy of supplementation and harvesting. Introducing two adult males and females annually will allow for sustainable harvesting to supplement wild populations. Additionally, this study identified 21 potential reintroduction zones in the Sundarbans and Terai regions of West Bengal, using environmental criteria that favor fishing cat occurrence. This work offers a comprehensive blueprint for applying the One Plan Approach to ensure the long-term survival of the fishing cat.

**Abstract:**

The One Plan Approach advocates for a hybrid species management framework, wherein captive-bred populations are considered metapopulations nested within a broader network of zoos and wild populations Additionally, the Opportunities to Thrive framework aims to enhance animal welfare by addressing the physiological, psychological, and emotional needs of captive individuals, thereby improving conservation outcomes. Here, we present an integrated framework for the conservation of a globally threatened wetland wild cat species, the fishing cat, by synthesizing optimal ex situ management practices and in situ conservation strategies. Further, we examined the genetic constitution of the founder population in a fishing cat captive breeding program that was recently initiated by the West Bengal Zoo Authority, India and conducted a population viability analysis to suggest how best to maintain the genetic diversity of the population. We found that the present genetic diversity of 56% and maximum carrying capacity of the captive population (30 individuals) can be maintained for more than 100 years with a combination of supplementation and harvesting. Keeping stochastic events in mind, the introduction of two adult males and females to the existing population each year will seamlessly allow the harvesting of two adult males and two adult females every alternate year to supplement wild populations. Further, we adopted the proposed integrated framework to delineate recommendations for the supplementation of wild populations in West Bengal. We used environmental criteria known to influence fishing cat occurrence to identify 21 potential reintroduction zones in the Sundarbans landscape and Terai region in northern West Bengal with habitable areas for the fishing cat that are larger than the maximum known species’ home range. Our study is timely and insightful because it provides a holistic blueprint for implementing the One Plan Approach in safeguarding a threatened species.

## 1. Introduction

We are currently witnessing the sixth mass extinction, characterized by an unprecedented decline in species populations and an escalation in anthropogenic threats [1,2,3]. The pivotal role of conservation efforts has never been more crucial, and the absence of concerted efforts would result in further deterioration in the status of species and higher incidences of extinction [4,5,6]. Management interventions to aid species persistence have become more critical than ever before [7]. This is because the combined effect of habitat loss and fragmentation is affecting species worldwide by creating small, localized wild populations that are highly susceptible to stochastic threats [8,9]. These circumstances necessitate an integrated approach to species conservation harnessing the strengths and opportunities offered by the ex situ domain to complement in situ conservation efforts [10,11,12] such as supplementing lost or declining wild populations periodically with zoo-bred “captive assurance populations” [13,14].

Zoos have been likened to the contemporary rendition of the mythological Noah’s Ark, with species conservation being one of their primary objectives since the 1990s [15]. Most zoos, however, have fallen short of fulfilling this mission, often confining animals without due consideration of their physiological and behavioral needs [16,17]. The original lofty idea of the zoo as Noah’s Ark quickly shipwrecked as several breeding programs ran into substantial problems. Zoos currently hold less than 15% of the threatened terrestrial vertebrates and close to none of the invertebrates [18]. Further, small sizes of founder populations render conservation breeding programs unsuccessful. Even though the initial target of maintaining 90% of genetic variability was reduced from a period of 200 years to 100 years, most of the captive breeding programs do not have sufficient space to house the required number of individuals to meet this objective. Subsequent to breeding in captivity, the prospects of reintroduction programs remain bleak for various reasons like financial considerations, as well as physiological and behavioral alterations in captive-bred animals [19,20]. However, a few successful reintroductions of captive-bred individuals for the rewilding and recovery of bottlenecked populations, in the recent past, are beacons of hope [21,22,23,24,25,26,27,28,29]. A hybrid form of species management was thus proposed, integrating in situ and ex situ conservation programs [30]. This approach envisions captive-bred populations as metapopulations nested within a broader network of zoos and wild populations [31].

The One Plan Approach for species conservation envisages a pivotal role of captive breeding and thereby animal welfare in fulfilling modern zoos’ conservation objectives [32,33]. The concept of animal welfare in zoos has evolved over the years, expanding its purview beyond the mere satisfaction of an animal’s physiological and psychological needs to encompass the animal’s capacity to experience both positive and negative emotions [34,35,36]. Notably, the San Diego Zoo Global has conceived a welfare framework known as Opportunities to Thrive comprising five components: (a) the provision of a strategically presented, well-balanced diet; (b) affording the animals the autonomy to self-maintain, i.e., freedom to avoid discomfort; (c) ensuring optimal health, i.e., freedom from disease; (d) facilitating the expression of species-typical behavior; and (e) providing choice and control, i.e., freedom to choose from multiple options available [33]. Positive indicators of animal welfare are known to be correlated with conventional markers, such as reductions in physiological indicators of stress (i.e., hormones), the incidence of health issues (i.e., body condition), and behavioral manifestations (e.g., successful breeding episodes), and more recently, also with complex indicators like the animal’s emotional state [33,37]. Conversely, poor welfare standards have resulted in stress-induced stereotypic behaviors and have concurrently reduced reproductive success and even maternal care for newborns, especially in carnivores [38].

In this manuscript, we explore opportunities to integrate ex situ strategies (including welfare protocols that will augment conservation outcomes) with in situ conservation planning for the globally threatened fishing cat (*Prionailurus viverrinus*). The fishing cat is a medium-sized, Indo-Malayan felid of global conservation concern and research priority [39,40]. It is morphologically adapted to persist in wetlands, and fish is the primary constituent of its otherwise eclectic diet [41,42]. It is also a member of the critically threatened freshwater biodiversity community facing significantly higher rates of decline compared to marine and forest ecosystems [43]. The cat has a patchy distribution range across south and southeast Asian countries including India, Nepal, Pakistan, Sri Lanka, Myanmar, Bangladesh, Vietnam, Java, and Cambodia [44,45,46]. Its presence in Sumatra, Lao PDR, Taiwan, China, peninsular Malaysia, and the west coast of India remains unclear [47,48,49]. It is classified as Vulnerable in the IUCN Red List related to its risk of global extinction, placed under Appendix 2 of CITES, and is a Schedule I species in the Wildlife Protection Act of India (1972). Its status is critically threatened in two range countries: Vietnam and Java (Indonesia), where the species has remained undetected for the past two decades. The alarming decline of wetland ecosystems, particularly in developing Asian economies, which encompass the entirety of the fishing cat’s global range, is the primary reason for the cat’s endangerment [45]. Their high occurrence in human-dominated landscapes also makes them vulnerable to hunting, retaliatory killing, and road kills [50,51,52,53]. The present circumstances necessitate the creation of a genetically diverse ex situ population to counter regional population decline and local extinction.

The specific objectives of the manuscript are the following: (a) to describe how the Opportunities to Thrive framework can be adopted for the maintenance of captive fishing cat populations, (b) to describe the best protocols to aid captive breeding, and (c) to propose a framework to integrate ex situ and in situ conservation approaches for global fishing cat conservation. Pertinently, the West Bengal Zoo Authority, India, has recently initiated a conservation breeding program for fishing cats. To consolidate this program, which is still in its infancy, we examined the founder population’s genetic status and used population viability analysis to suggest how the genetic diversity can be best maintained across the network of zoos in the state. This is the fourth objective of our study. Our final objective is to suggest how the captive bred population can best supplement the wild populations in West Bengal using our proposed integrated framework. The overall goal of this manuscript is thus to present an integrated framework for fishing cat conservation under the One Plan Approach and demonstrate its practicability through a case study.

## 2. Opportunities to Thrive Framework for Ex Situ Fishing Cat Populations

### 2.1. Strategically Placed, Well-Balanced Diet

The critical importance of tailoring captive species’ diets to align with their ethological needs is well appreciated in the literature [54]. Captive carnivores, for instance, are often given processed food, unlike the whole-carcass meals procured by their wild counterparts, which have behavioral, psychological, and physiological implications. This is especially true in zoological institutions within fishing cat range countries [personal observation]. Consuming processed food, for instance, limits the carnivore’s stomach extension, impacting sensations of fullness [55]. Consequently, these have increased foraging motivations, yet their confined environments preclude the expression of this natural behavior, resulting in chronic frustration, a marker of which is stereotypic behavior [56].

In the case of captive fishing cats, susceptibility to transitional cell carcinoma (TCC) has been linked to beef-rich diet [57,58]. Research has revealed notable differences in the fatty acid composition between fish- and meat-based diets given to captive fishing cats, with fish exhibiting lower omega-6 to omega-3 fatty acid ratios and higher levels of beneficial nutrients like potassium [59]. These factors help increase glomerular filtration rates (i.e., the rate of blood filtered by kidneys used as a test to assess kidney function), suggesting that fish-based diets have potential renal protective benefits [60,61]. Importantly, long-chain omega-3 polyunsaturated fatty acids, such as eicosapentaenoic and docosahexaenoic acids, found abundantly in fish oils, have anti-cancer properties [62,63]. Furthermore, dietary selenium increases cellular glutathione peroxidase activity [64,65]. Glutathione peroxidase has been shown to inhibit cyclooxygenase-2 (COX-2), an enzyme involved in inflammation and tumor activity [66]. Notably, fishing cats with TCC were found to overexpress COX-2 [58]. Therefore, a diet rich in selenium from fish may inhibit COX-2 function and potentially prevent TCC in fishing cats [59]. These findings have led to the adoption of a primarily fish-based dietary regime for captive fishing cats across zoological institutions.

In general, felids exhibit strong predatory instincts even when satiated, which can lead to abnormal behavior if captive environments are not sufficiently enriched [67,68,69]. The conventional feeding method for zoo felids, involving one daily meal of formulated food placed on the enclosure floor, disregards essential components of natural food acquisition in the wild—searching, locating, and capturing prey [69]. Small wild felids have two hunting strategies—(a) a sit-and-wait hunting strategy by which they wait for prey at a location in concealment and ambush it; and (b) actively flushing out prey [70]. Because small felids eat small prey [71], they typically engage in acquiring food multiple times a day [72]. While providing real predatory opportunities to captive felids might not be feasible, aspects of the predatory sequence can be simulated. This is particularly important in stimulating and nurturing curiosity in the surroundings, which is a prominent personality trait in wild cats [73].

Given this context, we asked for opinions from six experienced fishing cat keepers who are/were associated with the European Association of Zoos and Aquaria (EAZA) and the American Association of Zoos and Aquariums (AZA), and also consulted manuals published by zoological institutions to understand a) the diet of fishing cats in captivity, b) the presentation and placement of food, and c) how food can enrich the cat’s environment. This approach of consulting specialized keepers who work directly with the fishing cat was adopted due to the scarcity of scientific documentation on the species in ex situ facilities.

#### 2.1.1. Diet

Fish should account for 75% of the fishing cat’s diet. A minimum of three types of fish should be provided routinely, as this will provide the cats with a diverse array of nutritional compositions and profiles. Concentrations of protein can range from 40 to 75% (dry matter basis), and fat concentrations can range from 5 to 50%. However, fish fat can spoil quickly, leading to the rapid breakdown of vitamin E. In addition, enzymes such as thiaminase are very active in frozen fish and destroy thiamin fast. Therefore, commercially available supplements containing balanced proportions of thiamin and vitamin E should be provided to the cats. Due to the naturally elevated levels of vitamin A in fish, supplements formulated without additional vitamin A have been strongly recommended if a diet rich in fish is provided. The remaining 25% of the diet can be composed of a variety of items like rodents, chicken, rabbits, etc. It is recommended to sometimes provide whole carcasses of large rats and poultry and to provide 4–6 inches of an herbivore neck bone at least once a week. Multiple feeding opportunities should be offered throughout the day by splitting the daily food allowance between the feeding times. Set feeding times should be avoided as this will encourage stereotypic behavior. Overfeeding should be avoided as this can have a direct influence on the breeding ability of the cats. The female diet should be increased by 25–50% during pregnancy and by 50% during the last three weeks of pregnancy and lactation.

#### 2.1.2. Food Presentation and Placement

Shepherdson et al. [69] studied the effect of releasing live fish into pools with crevices for the fish to hide in, within a fishing cat enclosure. They observed a notable rise in active behavior related to predation, increasing from 0% to nearly 40% of their daily activity budget. Conversely, “less desirable” behavior like sleeping decreased from 66% to 21%, with the effects persisting for up to eight days following the introduction of live fish. This also increased behavioral diversity with increased utilization of the enclosure space, which was attributed to the increased locomotion between resting and hunting sites in an effort to explore the pool for more hunting opportunities. The extent and duration of these behavioral changes suggest the efficiency and effectiveness of presenting the fishing cat’s preferred prey in an environment that stimulates its curiosity and, more importantly, satiates the hardwired ethological need to explore for and locate prey. To stimulate predatory behavior, other meat should be given by providing animal carcasses, meat joints with bones, multiple small feedings, and by distributing, presenting, and concealing food in appropriate manners [74,75,76].

#### 2.1.3. Food as a Part of Environmental Enrichment

Food can be used to manipulate the captive cat’s environment daily so that the unpredictability of the wild can be replicated. Depending on the environmental conditions, items like frozen blood, frozen animal broth, gelatine made with blood/broth and water, pig ears, tuna, hardboiled eggs, housecat kibble, chicken feet, mussels, and insects can be hidden in different places within the living space of the captive animal to incite curiosity and searching. For example, food items can be placed inside thick pipes of various sizes, which could then be placed in different locations. Food items can be hung out of reach to stimulate repeated engagement.

### 2.2. Afford Self-Maintenance Opportunities

Captive animals need the opportunity to avoid discomfort such as inclement weather and overcrowding in zoos so that they can engage in essential self-maintenance behavior like grooming and bathing [33]. Cats, being naturally elusive, have a need for privacy. It is preferable to provide individual enclosures even for cats that co-exist harmoniously. This ensures the ability to separate them if needed due to incompatibility, injuries, food regulation, or for the introduction of animals for breeding purposes. Ground vegetation cover is essential to provide privacy to the cats, and it also provides the cats an opportunity to occasionally ingest vegetation as a digestion aid. A number of elevated platforms and aerial walkways should also be provided to the cats. Shy and nervous fishing cat individuals have been found to benefit from the provision of elevated platforms that they can use to gain vantage points for defense and observation, particularly during introductions. Additionally, a shallow pond with clean water should be provided for hydration and enrichment. Too many males in close proximity are likely to increase stress and tension between individuals and could result in the males ignoring females in favor of chasing away competitors.

### 2.3. Ensure Optimal Health

A strategic parasite control program is necessary as it aims to minimize reliance on anthelminthic treatment regimes by enhancing hygiene management practices alongside targeted anthelminthic use, thereby reducing anthelminthic resistance. Such a program facilitates the maintenance of the captive cat’s optimal health and decreases the chances of environmental contamination with parasites. Necessary steps to stop ectoparasite contamination or decrease parasitic load are the maintenance of hygienic feeding areas and the regular removal of feces from the area. It is important to ensure that each captive cat is administered the full dosage of the anthelminthic treatment, as inadequate dosage can lead to the survival and proliferation of resistant parasites, subsequently affecting both untreated and treated individuals. Apart from proper dosing, it is also recommended that treatment regimes rotate the administration of anthelminthic family medicines and injections. Post-anthelminthic treatment procedures also involve fecal examinations to verify the efficacy and address any treatment failures promptly. Additionally, screening new felids for disease and health status before their integration into breeding programs and vaccination against common feline viruses are essential measures to ensure the health of the captive population and the prevention of disease. Antibiotics should be used judiciously as these are known to reduce beneficial gut microbes and potentiate dysbiosis, i.e., disturbances to the typical gut microbiome, thus creating opportunities for secondary infections and facilitating antibiotic resistant bacterial strains [77,78].

### 2.4. Encourage Species-Typical Behaviour

Behavioral enrichment ideas involving scent are highly effective in increasing activity and reducing stereotypic behavior in fishing cats [79]. A variety of scents, including perfumes, cooking extracts, coffee grounds (in small quantities), tea (with caution regarding caffeine content), and spices such as oregano, basil, cloves, and catnip can stimulate species-typical behaviors. Toys also serve as an environmental enrichment option. Plastic or heavy tin containers filled with different substrates such as dirt, wood shavings, mulch, or leaves can promote behavioral diversity. Thick PVC pipes of various sizes can also be utilized to hide scented objects and encourage exploratory behaviors. Plant matter like lettuce, for instance, encourages shredding behavior in cats. Additionally, items like shed snakeskin and feathers offer further enrichment opportunities. Logs or stumps are another important enrichment option that encourage scratching behavior in these cats.

### 2.5. Provide Freedom to Choose from Multiple Options

Each female should ideally have access to a minimum of two nest boxes, although additional boxes are preferable. Different boxes will be preferred by different individuals, but they all have common factors, such as a small entrance that is just large enough for the cat to enter. The front of the box should be covered with vegetation to enhance seclusion. It is important not to disturb adult cats while they are in the nest boxes, especially if the female should associate this as a safe and secure area. Disturbing a female in the nest box is likely to result in her being unsettled and any disturbance during breeding could cause her to lose her kittens. Stressed females are also known to kill kittens, especially if it is their first litter.

## 3. Best Practices to Aid Captive Breeding

Fishing cats can be very aggressive and have been known to severely injure and even kill mates. Fishing cat breeding programs frequently encounter challenges stemming from the species’ discerning tendencies in partner selection. Experts unanimously suggest that cats should be housed in separate enclosures, but within close proximity, allowing initially for the exchange of olfactory stimuli (e.g., by keeping the fecal matter of females in the male’s enclosure and vice versa) and then graduating to visual stimuli (e.g., through a common window between enclosures). Generally, a long introduction over 4–5 weeks is preferred to reduce fatalities and injuries, as keepers can monitor and predict the behavior of the cats, and potential mates also have time to court each other as in the wild. This duration can be shortened (although to not less than 3 weeks) if active socialization without aggression is observed between the pairs. One of the advantages of such a gradual introduction process is the time allowed for courtship and the decreased potential of the male intimidating or injuring the female, which may not be keen to mate due to its oestrus state. Alternatively, introduction during the female’s oestrus period may increase the male’s receptivity and prevent female rejection. Often, such an introduction helps mitigate situations with mating-related conflicts, where the male is eager to mate with an unreceptive female. Close monitoring by keepers during the initial days post introduction is of utmost importance, with a contingency plan to separate the pairs if necessary. Additionally, both cats should receive extra food leading up to the introduction, with a substantial meal provided on the morning of the event, consisting of a preferred food type devoid of fur or bones. It is especially important for the male to have a large meal, as it will make him more sluggish and therefore less aggressive. Any food remains in the exhibit area should be removed immediately before the introduction takes place to prevent fights over food. Given the potential for severe aggression, captive cats should never be fed in the same enclosure, even if they are established as a pair, to mitigate the risk of fatal conflicts. Introductions can be carried out late afternoon/ early evening and onward when the cats are most active.

### 3.1. Ex Situ Conservation

In the proposed framework, the enhancement of welfare conditions for captive populations is prioritized. Subsequently, a comprehensive monitoring and evaluation process should be implemented to assess activities aimed at reducing stress, maintaining optimal health, fostering instinctive behavior, and improving the emotional well-being of the captive cats. Should the evaluation yield positive results, the zoo authorities may proceed with conservation breeding for this population. Conversely, a negative evaluation necessitates a thorough examination of gaps in practices and the implementation of corrective measures to sustain captive populations (Figure 1, Table 1). The primary criterion for engaging selected individuals in the founder population for captive breeding is their genetic robustness. Once achieved, we need to estimate the number of individuals necessary to preserve the genetic diversity. This genetically viable and robust population can then serve as a “captive assurance population” and can be periodically replenished with wild individuals. We suggest conditioning captive-bred individuals to in situ conditions through soft release set ups stationed inside habitat units where they are to be released, as against hard release events. This would also enable the gut microbial communities of these individuals to be rewired to persist in the wild, thereby providing the captive-bred individuals better chances of survival.

### 3.2. Integration with In Situ Conservation

Such contiguous habitats are expected to not only provide refuge to a viable fishing cat population but also represent lower anthropogenic threats from linear infrastructures and hunting. This approach provides the released individuals the opportunity to develop natural instincts, improve their chances of survival in the wild, and contribute to the replenishment of source populations (Figure 2). Ultimately, we hypothesize that this replenishment process will facilitate a “trickling out effect”, supporting the maintenance of meta populations across a wet landscape. Post-release, careful and prioritized monitoring of individuals is crucial to evaluate the effectiveness of the reintroduction process. Released individuals should be tracked with radio-collars and camera traps. The gut microbiomes of captive-bred individuals should also be monitored on priority post release. This will help in understanding whether the microbial community is being rewired to represent that of wild individuals, which has implications for relocation success. Such shifts in the post-release microbiome toward the wild incumbent microbiome is known to be crucial in supporting the survival of captive-bred carnivores in the wild [80].

## 4. Examining Genetic Robustness of Founder Populations and Requirements to Maintain Population Viability—Case Study from West Bengal

At the time of writing this document, there were 29 individuals in two captive locations in Kolkata, West Bengal—Zoological Garden, Alipore (ZGA), and Garchumuk Deer Park (GDP). This was close to the total carrying capacity of these facilities. As both the facilities are relatively close to each other and are within the biogeographic range of fishing cats in West Bengal, and since all the individuals are from the lower Gangetic floodplains, we considered them as a single captive population for all further analyses. Fishing cats are polygynous mammals and often reach sexual maturity at 9 to 12 months of age and continue to breed until they are 10 years old, with a maximum lifespan of 12 years in captivity. These cats give birth to one to four offspring, with a sex ratio of 1:1, twice a year. Litters with two to three offspring are more common. Our captive population had 14 males and 15 females, all in their reproductive age.

The fishing cat enclosure at GDP presently has a paddock area of 496 m^2^, divided into two equal parts by a wooden barrier 4 m high. The ZGA has a total area of 1047.34 m^2^ (display area is 409 m^2^ and off-display area is 638.281 m^2^) in which there are six night shelters. The night shelter cubicles ensure that individuals have enough space, with ample sunlight and ventilation. Landscaping in both the facilities is carried out such that the paddock area forms a wetland-like habitat within the enclosure with vegetation like *Typha elephantina*, *Saccharum spontaneum*, and *Colocasia esculenta* and shade trees such as *Vachellia nilotica*, *Mangifera indica*, and *Citrus maxima*. Fish are reared in artificial ponds within the enclosures, thereby encouraging captive individuals to hunt. Adults and kittens are fed daily with 750 g and 250 g of silver carp, respectively, and live snakeheads (*Ophiocephalus sps.*) are provided as feed enrichment. Live fish are known to improve the emotional well-being of fishing cats [69]. Moreover, natural prey helps establish a rich microbial community in the gut, contributing to the health of the captive population. The animals are regularly monitored for parasitic load and are dewormed once every three months. They are also given primary vaccinations with booster doses followed by annual vaccinations against canine distemper, feline pan leukopenia, rhinotracheitis, and rabies. Cleaning routines include the removal of litter and scats, 2% KMnO_4_ solution in foot baths at the entrances, the disinfection of night shelters with 0.5% Khorsolin TH solution, and rodent and other pest control.

We included genetic parameters as part of the population viability analysis (PVA) of the captive population. For this, we collected blood samples of all 29 individuals in EDTA-coated vacutainer tubes and isolated DNA with a Nucleospin Tissue Kit (Macherey-Nagel, Düren, Germany) according to manufacturer’s instructions except with longer incubation periods to obtain higher DNA yields. We checked the DNA concentration and purity with a NanoDrop ND-1000 (ThermoFisher Scientific, Waltham, MA, USA) and used approximately 50 ng/µL of DNA for all PCR assays. We screened a total of 22 microsatellite markers (Pbe01, Pbe02, Pbe03, Pbe05, Pbe06, Pbe09, Pbe10, Pbe13, Pbe15, Pbe26, Pbe31, Pbe32, F124, F115, F141, F42, F37, F53, Fca146, Fca391, Fca424, Fca441), which were reported to be polymorphic in domestic cat and leopard cat [81,82]. These markers were selected based on their reported polymorphism and amplicon sizes. PCR mix was always prepared in a UV-irradiated hood except for the addition of DNA. We set up all reactions in triplicates and every batch had a negative control to check for contamination if any. The PCR reaction mix of 15 µL comprised 1X Taq buffer (TaKaRa, Kusatsu, Japan), 250 µM of each dNTP (TaKaRa, Japan), 1X BSA (New England Biolabs, Ipswich, MA, USA), 3.5 pm of each forward and reverse primer, 0.75 U of Taq polymerase (TaKaRa Ex Taq Hot Start Version, TaKaRa, Japan), and ~50 ng/µL template DNA. We carried out PCR reactions in a Nexus Gradient Mastercycler (Eppendorf, Hamburg, Germany) with initial denaturation at 95 °C for 5 min, 40 cycles of denaturation at 95 °C for 30 s, annealing temperature varying for all loci from 50 °C to 64 °C for 30 s, extension at 72 °C for 30 s, followed by a final extension for 7 min at 72 °C. PCR success was checked electrophoretically in 2% agarose gel. Samples amplified in at least one out of every three replicates were analyzed further in triplicates by capillary electrophoresis in an ABI 3730 Genetic Analyser (Applied Biosystems, Waltham, MA, USA) along with the Genescan LIZ 500 size standard (Thermofisher Scientific, USA).

We scored alleles with GeneMapper 5.0 (Applied Biosystems, USA) and assembled the data in Microsoft Excel spreadsheets. We calculated the number of observed alleles, observed and expected heterozygosities, and polymorphic information content (PIC) of all loci with CERVUS 3.0.7 [83]. In total, 17 out of 22 microsatellite markers were polymorphic in fishing cat and were considered for further analysis. The mean expected and observed heterozygosities in this population were 0.555 and 0.557, respectively (Table 2). The genetic diversity of the fishing cat was found to be comparable to that of the leopard cat [82], but it is lesser than that of the domestic cat [81]. Deviations from the Hardy–Weinberg equilibrium (HWE) were determined with GENEPOP 4.7 (http://genepop.curtin.edu.au/) [84,85] with default values of Markov chain parameters, and the pairwise linkage (LD) disequilibrium was estimated using GENEPOP 4.7 (http://genepop.curtin.edu.au/) [84,85].

Only one marker, Pbe13, showed departure from the Hardy–Weinberg equilibrium. The closeness of the mean expected and observed heterozygosity values and the fact that the population falls in the HWE indicate that the alleles are assorted randomly, as expected. This also implies that the genetic composition of the population might not be significantly impacted by selection, inbreeding, or genetic drift. Eight pairs of microsatellite loci, out of 136 possible combinations, showed highly significant levels of association. We used ML-Relate [86] to calculate maximum likelihood estimates of relatedness (r) [87] and relationships from codominant genetic data. As the pedigree of the captive individuals was not known, we assessed the level of relatedness of each individual with every other individual in the population (Appendix A). We found 23 pairs of individuals with a high estimate of relatedness (>0.35), and ideally, these individuals should not be mated with each other in this captive breeding program.

Population viability analyses of the captive fishing cat population were simulated with Vortex v10 [88]. All simulations were projected for a period of 100 years with 100 iterations per simulation. We considered the species to go extinct when only one individual survived in the population.

### 4.1. Baseline Scenario

We used basic demographic indicators of the species to estimate the baseline scenario and assumed that this population will not be harvested from or supplemented, and it will also not be exposed to any life-threatening or catastrophic situation. We specified a mean distribution of 2.5 offsprings per female per brood, with a 75% chance of one brood per year and a 25% chance of two broods per year. The mortality rate of both males and females was considered to be 50% from age 0 to 1 year with a standard deviation (SD) of 10% due to environmental variation (EV), and the annual mortality rate of both males and females above 1 year was taken as 10% with an SD of 3% due to EV. We considered that 30% of the adult males in the population would monopolize all reproduction and could mate with one or two females in each season. Based on the average values derived from 40 distinct captive species of mammals, we considered a default inbreeding value of 6.29 lethal per diploid individual [89]. Simulations based on the above criteria showed that the captive fishing cat population will go extinct in 59 years (Figure 3a) with a deterministic growth rate (assuming no stochastic fluctuations, no inbreeding depression, no supplementation, and no harvest) of 25% (*r* = 0.25, λ = 1.28). The mean generation time for both females and males was predicted as 3.79 years, and the genetic diversity dropped from 55.5% to 18% in the 59th year before extinction.

### 4.2. Baseline with Catastrophe

We included the effect of catastrophe in the baseline scenario and analyzed the probability of population extinction due to disease outbreak. This is because an animal’s development or survival could be hindered by a number of local or global catastrophes even in confinement, where they are safeguarded and well cared for. As of yet, there has not been a catastrophic occurrence that has caused a significant decline in the captive fishing cat population. However, we cannot rule out future possibilities like epidemics of feline immunodeficiency virus [90], feline calicivirus [91], feline coronavirus [92], etc. We assumed a 20% chance of an epidemic occurring in any one year, and projected a 50% reduction in reproductive efficiency and a fall in survival efficiency by 10% during this event. In this scenario, the deterministic growth rate decreased further and dropped approximately 5.5% with respect to the baseline scenario (*r*= 0.195, λ = 1.21). With a disease outbreak, the population would decline faster and go extinct in 54 years with a genetic diversity of 23.5% (Figure 3b).

### 4.3. Baseline with Catastrophe and Harvesting

Here, we simulated the harvest of two adult males and two adult females each year starting from the 2nd year to the 100th year, over and above the catastrophe scenario described earlier. In this situation, the population would crash fast, and the genetic diversity would fall to 29.41% in the 24th year before the population went extinct (Figure 3c).

### 4.4. Baseline with Catastrophe and Supplementation

Captive populations require to be infused with unrelated individuals from the wild or from other zoos in order to preserve 90% of the genetic diversity obtained in the founder/wild population. In this scenario, we simulated that the captive population would be supplemented each year with two adult males and two adult females starting from the 2nd year to the 100th year. This is an ideal situation where the population would remain constant at the maximum carrying capacity of the facilities (~30 animals) and the genetic diversity would be 54.19% (Figure 3d).

### 4.5. Baseline with Catastrophe, Supplementation, and Harvesting

Finally, we combined the effects of catastrophe, supplementation, and harvesting to develop an ideal management strategy for the captive population. We simulated two possible ways of harvesting individuals from the population—one where we harvest two adult males and two adult females each year starting from the 2nd year to the 100th year, and a second where we harvest two adult males and two adult females once every two years starting from the 3rd year to the 100th year. While the genetic diversity remains approximately 54% in both the scenarios, even after 100 years, the number of individuals is better maintained in the second scenario (Figure 3e,f).

## 5. Supplementation of Wild Populations in West Bengal

In order to ensure the survival as well as the revival of rehabilitated fishing cats, it is necessary to conduct releases only in areas that can act as optimal habitats of the species. While wetland ecosystems are used and preferred by fishing cats [45], several anthropogenic factors threaten the ecological integrity and functioning of such habitats. To identify areas that can act as possible reintroduction zones, we generated a network of 1 km^2^ grids all across the state of West Bengal. This extent was chosen so as to align with the available spatial data on various environmental variables that were utilized to identify reintroduction zones, as well as to account for computational limitations in the analysis. Information on these environmental variables, as well as the criteria used to demarcate possible reintroduction zones, while taking into account fishing cat habitats and potential threats to their populations, is listed in Table 3 and explained in more detail below.

Among those possible reintroduction zones, we excluded areas that were at altitudes over 150 m above mean sea level, since fishing cats are known to live in lowland wetlands [40,94]. This was carried out using elevation data obtained using imagery from the Shuttle Radar Topography Mission’s (SRTM) digital elevation dataset [95]. Additionally, areas without the presence of wetlands were excluded, since the presence of wetland habitat is a key requirement for fishing cats to survive [45]. We gathered wetland spatial data from the Global Wetlands Map v.3 produced by a collaborative effort between the Centre for International Forestry Research (CIFOR) and the United States Forest Service [96]. However, such wetland habitat, when encroached upon by buildings and other human structures associated with settlements, can no longer act as prime fishing cat habitat. Geospatial data were obtained from the World Settlement Footprint, a product provided by the Earth Observation Center (EOC) of the German Aerospace Center, which provides high-resolution imagery of human settlements [97]. We removed points that coincided with areas recognized as settlements. Furthermore, many wetlands and wetland complexes have been converted to other land-use forms, such as agriculture and intensive commercial aquaculture. While aquaculture ponds provide a high concentration of fish prey, fishing cats are often killed in retaliation for fish predation in such ponds [51]. Spatial data on aquaculture were obtained from a dataset prepared by Ottinger et al. [93]. We gathered data on agriculture from the Landsat-derived Global Rainfed and Irrigated Cropland Product (LGRIP) [98].

Values for each environmental raster (listed in Table 3) at the centroid of each 1 km^2^ grid were used to represent the values for the respective grid itself, and using the above criteria, grids identified as optimal for fishing cat reintroduction were selected. Adjacent grids were then stitched together to demarcate reintroduction zones. Suitable locations for reintroducing captive-bred fishing cats need to be of a sizeable area of 30–150 km^2^. This is larger than known estimates of fishing cat home ranges [99,100], which is necessary as many mammal species that are released after translocation or captive breeding are known to exhibit exploratory behavior for up to several months in order to find a suitable habitat and resources [101]. The success of any reintroduction program will depend upon how few threats the released animal will encounter during this exploratory phase.

Using the criteria given in Table 3, we identified 33 possible reintroduction zones for fishing cats in West Bengal that are between 30 and 150 km^2^ (Figure 4). The average zone size was found to be 84.63 km^2^. Twelve zones were larger than 100 km^2^, and eight were smaller than 50 km^2^. Upon closer examination, using high-resolution Google Earth imagery, we found that 12 of the 33 zones overlapped with land uses that would be suboptimal for fishing cat reintroduction. Some zones fall within extents of large reservoirs, or along river courses (Figure 5), which are areas classified as wetlands using the remotely sensed CIFOR data. Since such habitats contain water bodies that are too deep or too wide for fishing cats, and these zones did not include significant terrestrial habitat along the watercourses, they were excluded. We also identified zones that appeared to coincide with high-density settlements and networks of aquaculture ponds that were not identified using the remotely sensed data used, as well as other intensive land-use types such as tea plantations and brick kilns. While fishing cats are known to persist in areas with settlements and aquaculture ponds [102], such areas are also prone to conflict, and act as ecological traps [103]. For the purposes of releasing captive-bred fishing cats, it is essential to select optimal locations where such threats are minimal, and therefore, the aforementioned zones were excluded. The remaining 21 reintroduction zones have an average size of 94.63 km^2^, which is larger than the average zone size when considering all 33 zones. This final set of optimal zones clusters around two main ecoregions in West Bengal. The Sundarbans is one major region optimal for fishing cat release, with 18 zones ranging from 36.014 to 150 km^2^ (Figure 6), while the other region is the North Bengal Terai–Duar landscape, which has three zones, the smallest being 39.814 km^2^, and the largest being 63.513 km^2^ (Figure 7).

## 6. Discussion

We adopted the Opportunities to Thrive framework and recommend that for the optimal physical, physiological, and emotional well-being of fishing cats, there should be (a) unpredictability in food acquisition as in natural circumstances, and specifically providing opportunities to hunt live fish; (b) practices to foster the much-celebrated curiosity of cats; (c) opportunities to maintain privacy; (d) freedom of choice; and (e) precautionary and preventive approaches for health maintenance. With respect to treatment regimes, we especially caution against the indiscriminate use of antibiotics because of their disruptive impact on gut microbial community, which has implications for the survival of the cats in the wild, post release [80].

The supplementation of wild populations must always be preceded by acclimatization in soft release centers, established inside target release sites, for a smooth and gradual adjustment of captive-bred individuals to the wild. For instance, captive-bred jaguars were moved to enclosures inside the Brazilian Pantanal from where they were released, following which it was seen that the released cats established their home ranges near the enclosure, hunted natural prey and reproduced successfully [24]. The continuous monitoring of captive-bred Iberian lynx, reintroduced in a similar way, showed their successful establishment in the wild after 5 years [25]. A slow transition and patient monitoring would also allow us to examine compositional shifts in the gut microbial community toward the wild incumbent microbiome post release—a process that is significant for the survival of captive-bred carnivores in the wild, as stated by Chong et al. [80]. Although not mentioned in the framework, resident fishing cat population abundances, population demographics, and prey densities at release sites should also be studied, if financial resources exist, as these factors could affect the released individual’s chances of survival.

The West Bengal Zoo Authority started a fishing cat breeding program that is the first of its kind, nested within the species’ distribution range. We found the genetic diversity of the founder population to be 56%. The presence of a genetically robust founder population is significant since it stands in stark contrast to fishing cat captive breeding programs in Europe and USA, where the homogeneous genetic constitution of the founder population has been an impediment to fishing cat breeding programs (personal communication; Linda Castaneda, 2020). The maximum carrying capacity of the captive population (30 individuals) in West Bengal can be maintained for more than 100 years with a combination of supplementation (i.e., the introduction of two adult males and females each year) and harvesting (i.e., the release of two adult males and two adult females to wild populations every alternate year). We identified 21 such potential release sites in West Bengal, with 18 sites in the Sundarbans delta region, i.e., in and around the Sundarbans Tiger Reserve and Sundarbans Biosphere Reserve, and the remaining three sites in the Terai region of the Himalayan foothills, and includes seasonally inundated grassland habitat in and around the Jaldapara Wildlife Sanctuary. In the context of the continuing and alarming decline of prime wetland habitat, it is likely that there will be growing disconnect between the populations in the Terai and the lower Gangetic floodplains. This is where the supplementation of wild populations will be crucial, if depauperate gene pools become a conservation concern in the near future.

Our approach to the selection of suitable release sites is based on the assumption that these sites need to have minimum anthropogenic footprint levels, especially urbanization, since captive-bred individuals are not accustomed to dangers associated with linear infrastructure and should also be safeguarded from possible negative interactions such as hunting and killing, which are known threats to wild fishing cat populations. Additionally, these will be less exposed to vectors from commensal species. Such an approach to minimize the exposure of captive-bred individuals to anthropogenic threats at their relocation sites was also adopted in the case of jaguars [24]. We further suggest that deworming medications should be administered to captive individuals as needed after proper investigations, which is against the current practice of administering them every three months in these captive breeding centers. The indiscriminate use of deworming medications can increase resistant parasitic strains in captive animals, and these strains can potentially be an added threat to wild populations once these individuals are released. Moreover, social surveys should be conducted in all the villages surrounding the larger habitat fragments in which fishing cats are recommended to be introduced. Such surveys should aim to understand the types of perceptions and attitudes that exist toward the fishing cat. The results could then be used to develop conservation strategies aimed to enhance the probability of successfully supplementing the wild population with captive bred individuals. For instance, if local people have an irrational fear of the fishing cat, sensitization programs should be taken up to dissipate fear and raise ecological knowledge of the fishing cat. On the other hand, if depredation on cultivated fishes and smaller livestock (like country goats) is an issue, participatory conservation programs to monitor, estimate, and replace actual loss should be initiated to increase tolerance and enable local residents to support conservation.

Despite our best efforts, some populations could be beyond genetic rescue due to extreme isolation and the presence of geriatric and/or non-reproductive individuals. It is here that evolving technologies in assisted reproduction can be used for the purpose of rescue and reinforcement [104]. Last but not the least, we see an opportunity for West Bengal’s “captive assurance population” to have relevance for supporting in situ conservation in other range countries like Vietnam. Fishing cats have not been detected in Vietnam for the past two decades, but suitable habitat is available, and therefore, re-wilding is a likely possibility, especially in the light of recent collaborations between the two countries to re-introduce tigers in Vietnam from the big cat’s population bank in India.

## 7. Conclusions

We elucidated how the physiological, psychological, and emotional wellness of captive fishing cats may be maintained by adopting the Opportunities to Thrive framework, and also identified parameters to monitor and evaluate the efficacy of these management recommendations. Ensuring the well-being of captive individuals will affect breeding programs that are envisioned to supplement in situ conservation programs under the One Plan Approach for holistic species management. Before any attempt is made to supplement wild populations, captive-bred individuals should be acclimatized in soft release centers, established inside the habitats marked for their release. The founder population of the fishing cat breeding program in West Bengal has a healthy genetic constitution. To maintain this genetic diversity, we recommend introducing two adult males and females each year into this captive metapopulation and releasing two adult males and two adult females every alternate year to supplement the wild. We identified 21 potential release sites in West Bengal, the populations within which could be supplemented in order to address potential loss in genetic connectivity due to rapid loss of wetland habitats.

## Figures and Tables

**Figure 1 animals-14-02770-f001:**
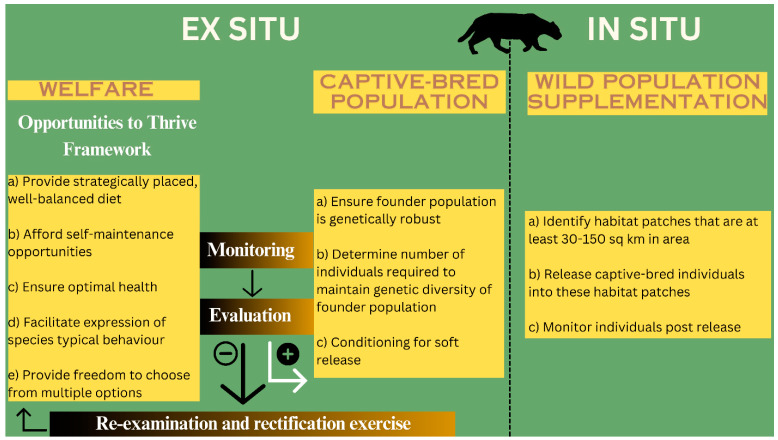
Diagrammatic representation of integrative framework for conservation of ex situ and in situ populations.

**Figure 2 animals-14-02770-f002:**
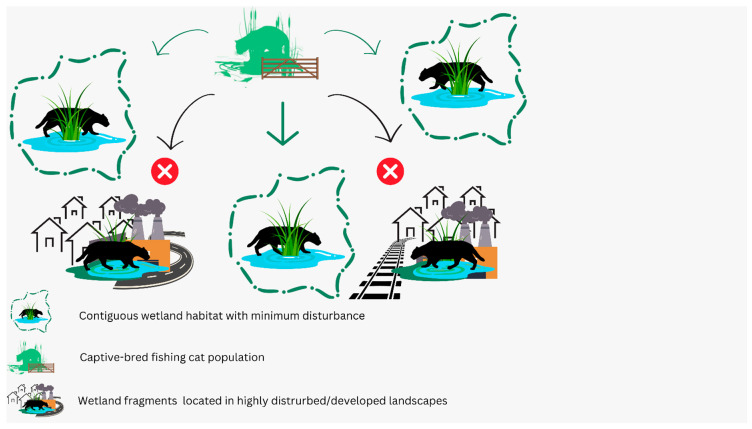
Framework to guide the process of the supplementation of wild populations, according to which captive-bred individuals should be released in locations with an intact wetland habitat and should not be released in locations that are situated in highly disturbed/developed areas.

**Figure 3 animals-14-02770-f003:**
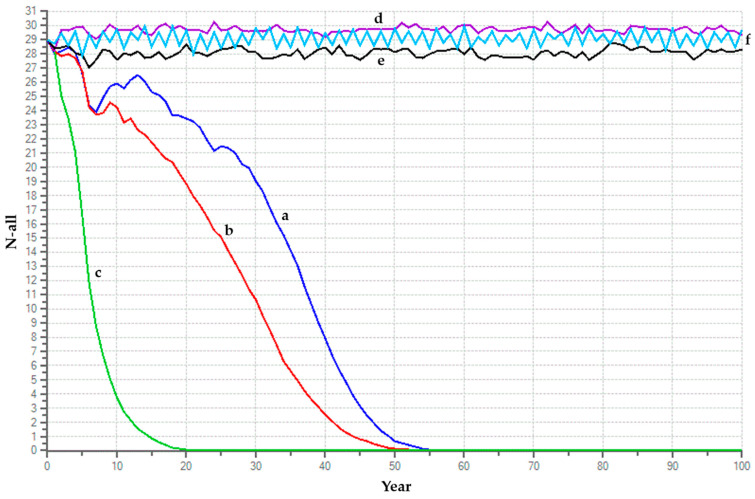
PVA graph indicating projected number of individuals (N) corresponding to simulated scenario for a period of 100 years. Scenarios are (a) baseline; (b) baseline with catastrophe; (c) baseline with catastrophe and harvesting; (d) baseline with catastrophe and supplementation; (e) baseline with catastrophe, harvesting, and supplementation scenario 1; (f) baseline with catastrophe, harvesting, and supplementation scenario 2.

**Figure 4 animals-14-02770-f004:**
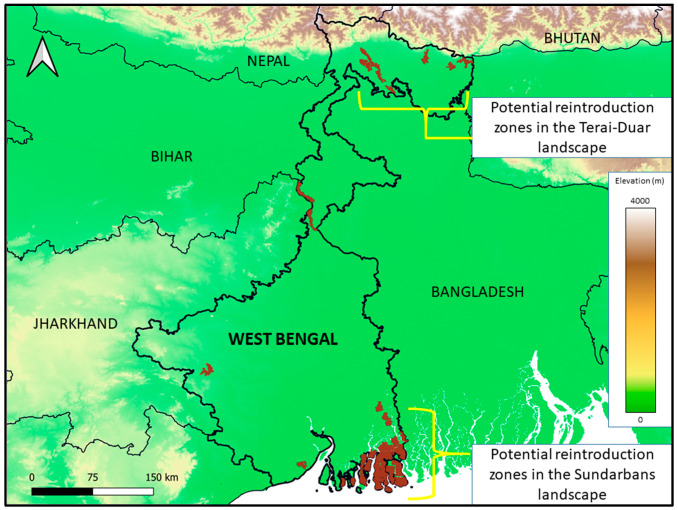
Elevation gradient within the state of West Bengal, with the 33 potential fishing cat reintroduction zones that are 30–150 km^2^ in size, including two key areas (Terai–Duar landscape in the north, and the Sundarbans in the south).

**Figure 5 animals-14-02770-f005:**
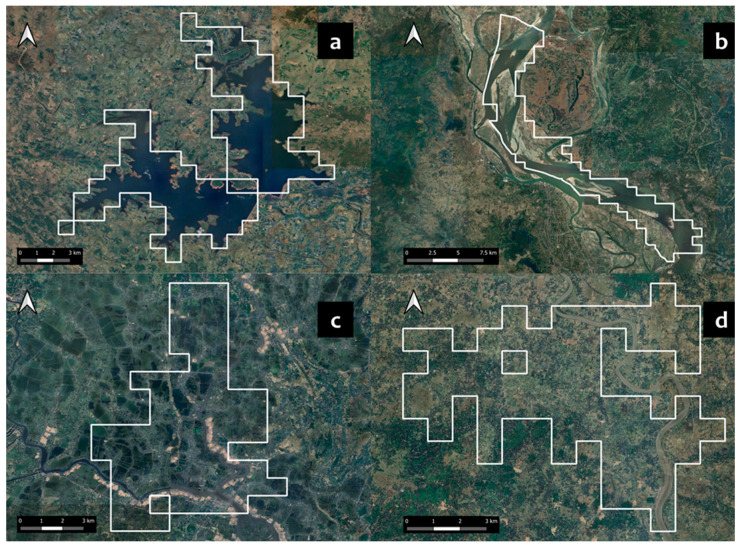
Examples of reintroduction zones that were excluded following the manual fine-tuning process. These zones were excluded because they fell within the extent of large reservoirs (**a**), large rivers (**b**), and intensive settlements and aquaculture that were not flagged while using remotely sensed data (**c**,**d**).

**Figure 6 animals-14-02770-f006:**
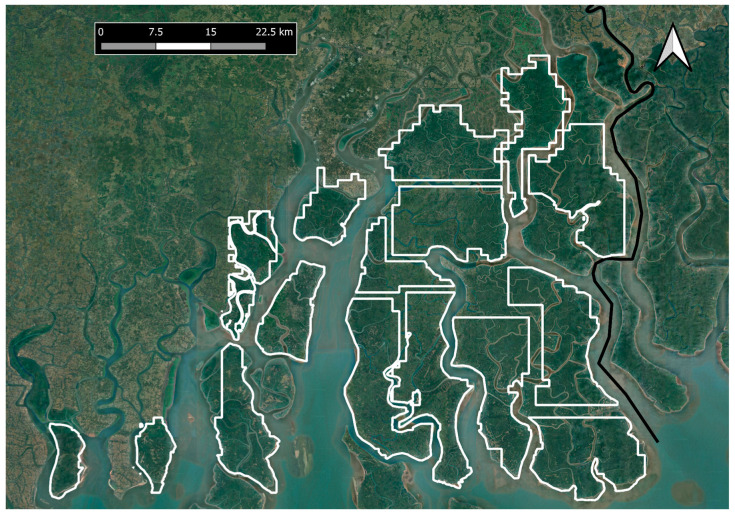
Possible reintroduction zones in the Sundarbans landscape. The black line on the map indicates the international border between India and Bangladesh.

**Figure 7 animals-14-02770-f007:**
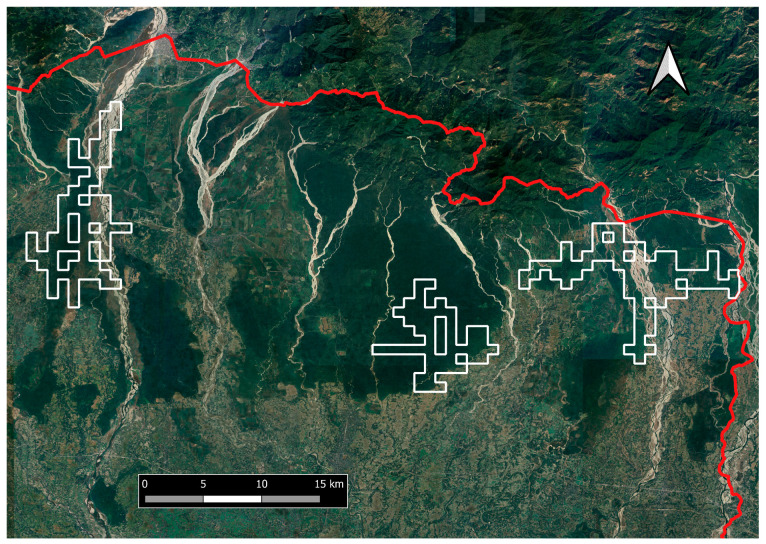
Possible reintroduction zones in the North Bengal Terai–Duar landscape. The red line on the map indicates the international border between India and Bhutan.

**Table 1 animals-14-02770-t001:** Indicators used to evaluate the implementation of the Opportunities to Thrive framework.

Indicators to Evaluate the Implementation of the Opportunities to Thrive Framework	Example of Indicators to Evaluate Interventions Taken under the Opportunities to Thrive Framework
(a) Physiological health	Reduction in stress hormones.
(b) Physical health	Improved body condition or maintenance of optimal body conditions.
(c) Behavioral diversity	Decrease in negative behaviors such as weight loss, lack of appetite, excessive pacing, self-mutilation, constant hiding Increase in positive markers such as an overall increase in behavioral diversity such as exploring, scent marking, resting, eating, engaging with the environment, successful breeding, etc.
(d) Emotional health	Positive vocal/postural signals

**Table 2 animals-14-02770-t002:** Statistical indices of genetic diversity of the captive fishing cat population in West Bengal obtained with 17 polymorphic microsatellite loci.

Locus	No. of Alleles	Observed Heterozygosity	Expected Heterozygosity	PIC	HWE
Pbe02	7	0.759	0.819	0.779	0.1798
Pbe03	4	0.571	0.513	0.451	0.9375
Pbe06	3	0.292	0.260	0.231	1.0000
Pbe10	4	0.318	0.290	0.269	1.0000
Pbe13	5	0.481	0.760	0.707	0.0136
Pbe15	4	0.217	0.205	0.192	1.0000
Pbe26	4	0.538	0.479	0.420	0.8748
Pbe31	3	0.286	0.254	0.226	1.0000
F124	7	0.759	0.737	0.681	0.2044
F115	10	0.828	0.867	0.836	0.2409
F141	3	0.655	0.656	0.570	0.2013
F42	6	0.667	0.740	0.685	0.5063
F37	3	0.556	0.451	0.360	0.5521
F53	6	0.800	0.774	0.713	0.1607
Fca146	2	0.143	0.136	0.124	1.0000
Fca424	4	0.773	0.760	0.695	0.9604
Fca441	4	0.826	0.737	0.668	0.8910
Mean	4.647	0.557	0.555	0.506	

PIC—Polymorphic Information Content; HWE—Hardy-Weinberg Equilibrium.

**Table 3 animals-14-02770-t003:** Environmental variables, their sources, and the criteria used to identify apt habitats for fishing cat reintroduction.

Environmental Variable	Source	Criterion
Wetland	CIFOR	Presence of wetland (any type)
Elevation	SRTM	<150 m
Urbanisation	World Settlement Footprint	Absence of settlement
Aquaculture	Ottinger et al. [93]	Absence of aquaculture pond
Irrigated Agriculture	LGRIP	Absence of agriculture (irrigation type)

## Data Availability

The raw data supporting the conclusions of this article will be made available by the authors upon request.

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
