# Peer review of "Making Noah’s Ark Work for Fishing Cat Conservation: A Blueprint for Connecting Populations across an Interactive Wild Ex Situ Spectrum"

_animals, 2024, doi:10.3390/ani14192770_

Round 1

Reviewer 1 Report

Comments and Suggestions for Authors

This very well-written paper seeks to propose a framework for restoring wild fishing cat populations from captive founders.  The topic has wide appeal based on the subject (fishing cats) and methods (ex-situ, in-situ reintroduction).

The introduction clearly sets out the context for the paper. Relevant existing work is presented, and the case for the work to be undertaken is clear. That said, I wonder if this is two papers rather than one.

In the final paragraph of the Introduction, the authors note the 5 objectives of the paper. This makes for quite a busy paper that is arguably trying to do too much.  Although these Objectives are related to each other, Objectives a) and b) (and potentially c) could be a single paper that presents a case for in-situ and ex-situ reintroduction and includes how this might be managed in principle.  A second paper could then apply this to a real-life case - that of West Bengal Zoo.  This would mean that the Conclusions section - which is currently quite brief - could be developed further for each paper.

Throughout the paper, I found I had to keep flipping back to page 3 to work out which Objective was being served in each section. Although all five Objectives are served, it can feel a little superficial at times.  For example, the in-situ conservation strategy should include an assessment of attitudes and behaviours towards fishing cats, the history of HWC, programmes to mitigate, and the availability of prey. This is not mentioned here.  Hence, splitting this paper into two would facilitate a richer engagement with the Objectives.

 I do not have sufficient expertise to assess the work on genetic robustness.

As noted above, the conclusions drawn are fair but under-developed. I think this is a consequence of trying to do too much in one paper. 

Reviewer 2 Report

Comments and Suggestions for Authors

I enjoyed reading the manuscript " Making Noah’s Ark Work for Fishing Cat Conservation: a blue-print to Connect Populations across Interactive wild-ex-situ Spectrum ". The manuscript has a comprehensive literature review, a clear framework for welfare improvement, and an integration of conservation goals. The use of 22 microsatellite markers and the robust statistical analysis performed are commendable. It has the following areas for improvement:

1.     The manuscript occasionally shifts between different terms for the same concept (e.g., “ex-situ” vs. “captive”), which might confuse readers. Consistently using the same terminology throughout the manuscript would enhance clarity.

2.     Some technical terms (e.g., "glomerular filtration rates") are not fully explained. Providing brief definitions or explanations would make the content more accessible to a broader audience.

3.     It could benefit from a clearer discussion on the methodology used to gather and analyze expert opinions. For example, how were the opinions from the zoo associations (EAZA, AZA) collected? Detailing the process and number of participants would add accuracy to the study and help in assessing the validity of the conclusions.

4.     The document is well-organized, but consider adding subheadings within sections to improve readability. For example, under “Population Viability Analysis,” it is better to have subheadings like “Sample Collection,” “DNA Isolation,” and “PCR Assays.”

5.     While the manuscript addresses welfare concerns, it could include a more in-depth discussion of the ethical implications of captive breeding and reintroduction programs. This could involve exploring the potential risks to both the animals and the ecosystems they are released into, as well as the ethical responsibilities of the institutions involved.

6.     The manuscript reports observed and expected heterozygosity values but does not fully explain the implications of these findings.

7.     The significance of the deviations from Hardy-Weinberg equilibrium should be elaborated more.

8.     The baseline scenario results indicate a population decline to extinction within 59 years. However, the assumptions made for the demographic parameters, such as the mortality rates and breeding probabilities, could be better justified. Additionally, in lines 450-452: the assumption 20% annual probability of a disease outbreak is quite high. Justifying this choice with references or providing sensitivity analysis to explore how varying this probability affects the outcomes would strengthen the scenario's credibility. 

Overall, the manuscript presents a well-structured and comprehensive framework for improving the welfare of captive fishing cats, with a strong emphasis on conservation is a valuable contribution to the field of conservation biology, and conservation genetics.

Reviewer 3 Report

Comments and Suggestions for Authors

The study provides two frameworks for populations of the endangered species of fishing cats, one a "planned approach" and the other an "Opportunities to Thrive". The paper is informative and voluminous, and is of interest to zoo populations of this species. Providing an important role in the conservation of populations of the endangered species of fishing cats, these deserve publication.

I only have a couple of comments and suggestions, and I hope these will be easy to address.

1.      Punctuation is missing in some of the language, affecting the reading experience

2.      Keywords should not be duplicated in the title

3.      The language in the introduction section is lengthy and does not do a good job of highlighting the main highlights of this study. The foreword section needs to be reorganized by the author to make the article more concise.

4.      Line 92- 94. I don't think this sentence should be here, it should summaries the current situation of the ZGA and the GDP zoo, thus over-introducing the fishing cats and making the logic flow.

5.      Line 105 L should cite IUCN Current endangerment of fishing cats

6.      Line 165-170 and (B) both refer to food presentation and placement, and it is suggested that the summary of the paragraph be made more concise to enhance the readability of the article. The article as a whole should be carefully revised to address this issue.

7.      Lines 275-276 should be supported by citations to relevant literature.

8.      Indicators to assess the implementation of theOpportunities framework. It is recommended to use "+" for assessment, e.g. "++++" for optimal body conditions, "++" for healthy body condition, "+" for sub-healthy body condition and "-" for ill health body condition. ", Sub-healthy body conditions for "+", ill health body condition for "-".

9.      Line 344 suggests that it would be clearer to have subparagraphs in the chapters, e.g., overview of the study site, methodology, results, etc.

10.  Whether the reagent is followed by (place name, country) or (company, country) should be harmonized.

11.  The sentence introducing the fishing cat should not appear in the results section. Line 423-426.

12.  It is recommended that figure 3 be labelled a, b, c or that the colors of a, b, c be indicated on the figure note.

13.  Figure 4 needs to have a compass, altitude, name of the area, and a representation of what the color yellow means.

14.  Figure 5 It is recommended that the labelling should not be top left and right. a, b, c, d is clearer. The compass should be the same size

15.  The article is full of small problems that need to be checked and revised by the author.

Comments on the Quality of English Language

Moderate editing of English language required.

Reviewer 4 Report

Comments and Suggestions for Authors

Initially, this reviewer would like to extend compliments to the authors for their remarkable efforts in the conservation of the fishing cat. The comments and criticisms presented below are made with the best intentions, aiming to contribute to the enhancement of this manuscript.

1. Brief Summary

The manuscript aims to present an integrated framework for the conservation of the globally threatened fishing cat by synthesizing optimal ex situ management practices with in situ conservation strategies. It highlights the One Plan Approach and Opportunities to Thrive framework as pivotal in enhancing both captive and wild populations. The paper's strengths lie in its comprehensive approach to species management, addressing genetic diversity, and proposing practical conservation strategies that could potentially be applied to similar species facing conservation challenges.

2. General Comments

Introduction: The introduction relies heavily on references that are over ten years old, which may not accurately reflect the current state of conservation science. It would be beneficial to incorporate more recent literature to provide a contemporary context.

Objective Clarity: One of the specific objectives stated was the “description of the best protocols to aid captive breeding.” However, the manuscript does not adequately address this objective. There is a lack of substantial content on captive breeding protocols.

Incorporation of Recent Frameworks:

- The Conservationist's Toolkit – see https://doi.org/10.1016/j.biocon.2023.110345
Consider integrating this framework, which offers a conceptual approach to both in situ and ex situ conservation strategies. This toolkit can guide the development of strategies that consider ecological, social, and management variables, crucial for connecting and maintaining fishing cat populations across different environments.

- One Conservation concept – See https://doi.org/10.1590/1984-3143-AR2021-0024
In addition to the One Plan Approach, the manuscript should incorporate the One Conservation concept, which emphasizes connecting conservation actors and bridging gaps between wild and captive population management. This concept highlights the importance of involving parties traditionally not engaged in conservation efforts. See also https://doi.org/10.3389/fvets.2022.897404 (about the need for protocols) and https://doi.org/10.1016/j.therwi.2023.100024 (this concept in practice)

Assisted Reproduction: The authors should consider discussing assisted reproduction as a critical tool in conservation programs. An example of its application in felid species can be found in https://doi.org/10.1016/j.therwi.2024.100070

3. Specific Comments

Lines 65-67: The statement regarding the bleak prospects of reintroduction programs is outdated. Numerous successful reintroduction cases exist and should be referenced to provide a balanced view. Consider including recent examples from the literature. Suggested references include:
https://doi.org/10.1017/S0030605320000460
https://doi.org/10.1002/zoo.21577
https://doi.org/10.3390/d13050198
https://doi.org/10.3390/d16020080
https://doi.org/10.1038/s41598-021-93673-z
https://doi.org/10.1371/journal.pone.0249860
https://doi.org/10.1016/j.gecco.2021.e01860
https://doi.org/10.3390/ani12223158

For a successful case in Iberá, Argentina:
https://doi.org/10.1111/csp2.258
https://doi.org/10.1017/9781108638142.024
https://doi.org/10.4324/9781003097822-19

Line 105: Correct the phrasing to “classified as 'Vulnerable' related to its risk of global extinction.”

Line 130: The term "Recent research" should be used cautiously. Generally, research conducted within the last 5 years is considered recent, particularly in rapidly advancing fields. The cited work from 2017, originally presented in 2013, does not qualify as recent.

Lines 131-134: The claim regarding carnivores being fed processed food without consideration of their needs is outdated. Current practices have evolved significantly. Current manuals provide well-developed recommendations. [See: AZA Animal Care Manuals https://www.aza.org/animal-care-manuals , EAZA Best Practice Guidelines https://www.eaza.net/conservation/programmes/#BPG , National Animal Interest Alliance https://nagonline.net/ ]

Lines 135-137: The assertion about chronic frustration due to limited environments may not apply to all institutions. Consider revising to acknowledge advancements in enrichment practices, reviewing more recent literature, such as https://doi.org/10.3390/ani14152223

Line 139-140: The claim linking TCC to commercial carnivore diets is not supported by the references cited. Sutherland-Smith et al. did not establish this link, and Landolfi and Terio (2006) only suggested the need for further investigation into potential dietary influences. The only study suggesting a dietary influence is Marshall et al. (2013 – see VME-23 at https://vet.osu.edu/sites/default/files/documents/2013BookOfAbstracts.pdf), which did not involve commercial diets.

Lines 208-216: This section lacks references. Including recent studies on environmental enrichment would strengthen the claims made. Suggestion https://doi.org/10.3390/ani13081277

Lines 226-228 and 230-232: These statements require supporting references to validate the claims made about self-maintenance and social dynamics.

Line 276: The statement about expert consensus on housing arrangements lacks a citation. Providing a reference would enhance the credibility of this claim.

Lines 317-318: While replenishment with wild individuals is discussed, the manuscript would benefit from exploring assisted reproduction and biotechnologies as alternative strategies to enhance genetic diversity without removing individuals from the wild. Refer to the One Conservation concept.

Line 361: Clarify the purpose of using live prey if the animals are not intended for reintroduction. WAZA guidelines recommend avoiding live prey unless essential for the predator's health and welfare. Behavioral enrichment can often be achieved through alternative methods. See https://www.waza.org/wp-content/uploads/2019/03/WAZA-Animal-Welfare-Strategy-2015_Landscape.pdf

Lines 362-363: If parasitic load is regularly monitored, why is deworming conducted every three months rather than as needed? Indiscriminate use of dewormers can increase resistance.

Line 363: What is the prescribed vaccination schedule?

By addressing these comments, the manuscript can be significantly strengthened, providing a more comprehensive and up-to-date contribution.

Round 2

Reviewer 1 Report

Comments and Suggestions for Authors

Thanks to the authors' efforts, this is a much-improved paper. Although it is still a 'busy' paper, it focuses more on its aims and objectives. As I noted in my initial review, I am not sufficiently well-versed in genetics to comment on this aspect of the work, but I am now satisfied that the comments made by reviewers have been addressed in the revised version. 

Reviewer 2 Report

Comments and Suggestions for Authors

No comments

Author Response

There are no comments.

Reviewer 3 Report

Comments and Suggestions for Authors

Dear Author.

I do not accept some of your explanations and the changes you have made do not address the issues I have raised.

Add methodology, results, etc. as headings to the large integrated paragraphs.

  Fig 4 Pink area enlarged and described separately next to the diagram, preferably in color

No DOI in some of the reference.

Comments on the Quality of English Language

Minor language changes required.

Reviewer 4 Report

Comments and Suggestions for Authors

This Reviewer commends the authors once again for their dedicated efforts in the conservation of the fishing cat. The critiques provided are intended to enrich the manuscript.

The authors' deep knowledge of the species and the work conducted is acknowledged. The suggestions previously made aim to incorporate modern ideas and perspectives into conservation efforts.

Regarding the concepts of "The Conservationist's Toolkit" and "One Conservation," the authors' stance is understood. However, incorporating these into the discussion as a future step in the species' conservation actions could enhance the manuscript without requiring significant changes. These contemporary approaches would only benefit the work. While this Reviewer respects the authors' decision not to adopt these views, this choice does not affect the recommendation for the manuscript's publication.

In relation to the statements on reproductive biotechnologies, this Reviewer understands the points raised. Nevertheless, it is important to emphasize the significance of these biotechnologies in the conservation of endangered species. With some knowledge of the current scenario in India, it is observed that, while the government invests significantly in biotechnologies for livestock, investment in wild animals is negligible. However, the private sector has been active in the assisted reproduction of wild species. Within the "One Conservation" concept, active private sector participation is crucial, representing an opportunity for the fishing cat. The Vantara Foundation, for instance, could be a potential partner.

Understanding the authors' arguments, it is recommended to address assisted reproduction not in the present, but as a future perspective. It is suggested to create a subsection before the conclusions, titled “Looking to the Future,” to discuss assisted reproduction as a future strategy for the species.

Concerning the statement, “Many zoos, including ones in India, still provide processed food and therefore we think that this is still mention-worthy and should be cautioned against,” it would be pertinent to specify in the manuscript that this practice is common in the region where the fishing cat occurs. The term “Many Zoos” is generic, as this practice may not be usual in other continents. Thus, it is important for the authors to clarify this issue in the text.

For the sections previously highlighted for the lack of references, where the authors responded, “All information provided here were received from expert fishing cat keepers in zoos,” it is suggested that the authors clarify (in lines 214-222; 232-234; 236-238) that, due to the scarcity of scientific documentation on the species, the data are based on information from specialized keepers who work directly with the fishing cat.

In line 380 (formerly 361), this Reviewer appreciates the information provided by the authors in the response letter, recognizing its importance in the text. The inclusion of justifications for the use of live fish, such as “Live fish is known to improve the emotional well-being of fishing cats. Moreover, more natural prey helps establish a rich microbial community in the gut, contributing to the health of the captive population,” is highly recommended.

Finally, the CONCLUSIONS section is excessively long and does not comply with the journal's "Instructions for Authors," which requests “Conclusions: This section is mandatory, with one or two paragraphs to conclude the main text.” Therefore, it is necessary to adjust this section, transferring more appropriate parts to the DISCUSSION.
